# B Cells and Tertiary Lymphoid Structures Influence Survival in Lung Cancer Patients with Resectable Tumors

**DOI:** 10.3390/cancers12092644

**Published:** 2020-09-16

**Authors:** Jun Tang, Daniel Ramis-Cabrer, Víctor Curull, Xuejie Wang, Mercé Mateu-Jiménez, Lara Pijuan, Xavier Duran, Liyun Qin, Alberto Rodríguez-Fuster, Rafael Aguiló, Esther Barreiro

**Affiliations:** 1Pulmonology Department, Lung Cancer and Muscle Research Group, Hospital del Mar-IMIM, Parc de Salut Mar, Health and Experimental Sciences Department (CEXS), Universitat Pompeu Fabra (UPF), Medical School, Universitat Autònoma de Barcelona, Parc de Recerca Biomèdica de Barcelona (PRBB), 08003 Barcelona, Spain; jun.tang2@e-campus.uab.cat (J.T.); daniel.ramis@ssib.es (D.R.-C.); VCURULL@PARCDESALUTMAR.CAT (V.C.); Xuejie.Wang@e-campus.uab.cat (X.W.); merce.x.mateu@gsk.com (M.M.-J.); liyun.qin@e-campus.uab.cat (L.Q.); 2Centro de Investigación en Red de Enfermedades Respiratorias (CIBERES), Instituto de Salud Carlos III (ISCIII), 08003 Barcelona, Spain; 3Pathology Department, Hospital del Mar-IMIM, Parc de Salut Mar, 08003 Barcelona, Spain; LPIJUAN@PARCDESALUTMAR.CAT; 4Scientific, Statistics, and Technical Department, Hospital del Mar-IMIM, Parc de Salut Mar, 08003 Barcelona, Spain; xduran@imim.es; 5Thoracic Surgery Department, Hospital del Mar-IMIM, Parc de Salut Mar, 08003 Barcelona, Spain; ARodriguezFuster@parcdesalutmar.cat (A.R.-F.); RAGUILO@PARCDESALUTMAR.CAT (R.A.)

**Keywords:** lung cancer, chronic respiratory diseases, tertiary lymphoid structures, B cells, overall survival

## Abstract

**Simple Summary:**

Nowadays, humans still die of lung cancer (LC), a disease mainly related to cigarette smoking (CS). Smokers also develop chronic bronchitis, namely chronic obstructive pulmonary disease (COPD). Environmental factors and a natural predisposition from the patients’ sides may render them more prone to develop tumors derived from CS. Thus, a great number of patients may suffer from chronic bronchitis and LC simultaneously. Chronic respiratory diseases are also important risks factors for LC. The immune system, among other biological mechanisms, protect our cells from infections and cancer development. Several immune structures and cells may be altered in the tumors of patients with COPD as opposed to lung tumors of patients with no underlying respiratory disease. A total of 133 patients with LC participated in the study: 93 with underlying COPD. Several structures (tertiary lymphoid structures, TLS) and T and B lymphocytes were analyzed in the lung tumor and non-tumor areas (specimens obtained during surgical extirpation of the tumors). We found that in LC patients with COPD, compared to those without it, fewer numbers of TLSs and B cells were detected, and those patients died significantly earlier. These results have implications in the diagnosis and treatment options of lung tumors in patients with underlying respiratory diseases.

**Abstract:**

Immune profile of B and T cells and tertiary lymphoid structures (TLSs) may differ in tumors of lung cancer (LC) patients with/without chronic obstructive pulmonary disease (COPD), and may also influence patient survival. We sought to analyze: (1) TLSs, germinal centers (GCs), B and T cells, and (2) associations of the immune biomarkers with the patients’ 10-year overall survival (OS). TLSs (numbers and area), B [cluster of differentiation (CD) 20], and T (CD3), and GCs cells were identified in both tumor and non-tumor specimens (thoracotomy) from 90 LC-COPD patients and 43 LC-only patients. Ten-year OS was analyzed in the patients. Immune profile in tumors of LC-COPD versus LC: TLS numbers and areas significantly decreased in tumors of LC-COPD compared to LC patients. No significant differences were observed in tumors between LC-COPD and LC patients for B or T cells. Immune profile in tumors versus non-tumor specimens: TLS areas and B cells significantly increased, T cells significantly decreased in tumors of both LC and LC-COPD patients. Survival: in LC-COPD patients: greater area of TLSs and proportion of B cells were associated with longer survival rates. The immune tumor microenvironment differs in patients with underlying COPD and these different phenotypes may eventually impact the response to immunotherapy in patients with LC.

## 1. Introduction

Lung cancer (LC) is still the most common cause of death worldwide [1,2,3,4,5], accounting for almost one-third of deaths in certain geographical areas [6]. Chronic respiratory diseases, such as chronic obstructive pulmonary disease (COPD), which is also a highly prevalent condition in certain regions, has been consistently associated with LC incidence [7,8]. Airway obstruction and emphysema are, indeed, important risk factors for LC [7,8]. Assessment of the biological mechanisms that render patients with chronic lung diseases more susceptible to LC development remains to be fully elucidated.

In the process of tumorigenesis, inflammatory events interact with several cellular mechanisms such as angiogenesis, apoptosis, cell repair, and distant metastasis, which are promoted by cytokines and growth factors [9,10]. Tumor microenvironment is also crucial in the development of LC, its progression, and response to therapy in clinical settings. Immune surveillance is relevant to the microenvironment of the tumor lesions as it may interfere with disease progression. Antitumor effects are exerted by T helper (Th) 1lymphocytes, whereas Th2 cells may inhibit the host immune system, thus, favoring tumor development and growth [11]. LC relapse and response to immunotherapy also rely on the balance between Th1 and Th2 immune phenotype [9,12,13,14]. Moreover, Th1 and Th2 immune response may vary in patients with underlying respiratory diseases [15,16]. In accordance, a previous study clearly demonstrated that Th1 cytokines were predominant in the tumors of patients with LC and underlying COPD, suggesting that these patients exhibited a greater inflammatory profile that might be beneficial in response to certain therapies [10]. The specific pattern of immune cells present in lung tumor specimens of patients with LC and COPD remains unanswered.

Tertiary lymphoid structures (TLSs), which share identical characteristics to lymph nodes, are encountered in inflamed and infected tissues and in tumors. They are characterized by the presence of a T cell area, germinal centers, and proliferating B cells among other structures [17,18,19]. In COPD patients, a greater number of TLSs were demonstrated in lung tissues [20]. Whether TLSs may be involved in LC development in patients with COPD is still debatable. Our hypothesis was that tumor microenvironment, as assessed by the profile of TLSs and the number of B and T cells, may differ in tumors of patients with underlying COPD compared to those without this disease, and these differences may also influence patient survival. Hence, our objectives were that in lung tumors and non-tumor specimens of LC patients with and without COPD: (1) TLSs, germinal centers (GCs), and B and T cells were explored, and (2) associations of these immune biomarkers with the patients’ 10-year overall survival (OS) were assessed. All of the patients were clinically followed up, to a maximum period of 10 years, for the analyses of the survival.

## 2. Results

### 2.1. Clinical Characteristics of the Study Patients

Table 1 describes all clinical and functional features of both LC and LC-COPD patients. The number of LC-COPD patients was greater than that of LC only patients (two-fold), with predominance of male patients. No significant differences were seen in age or body mass index (BMI) between LC-COPD and LC patients. Expectedly, the percentage of ex-smokers and the number of packs–year were significantly higher in LC-COPD patients than LC patients, while the number of never-smokers was significantly greater in the latter group (Table 1). As expected, lung functional parameters were significantly lower in LC-COPD patients than in LC patients (Table 1). The majority (91%) of LC-COPD patients were in Global Initiative for Chronic Obstructive Pulmonary Disease (GOLD) I and II stages. In addition, no differences were observed in tumor, node, and metastasis (TNM) staging or histological subtypes between both groups of patients. Compared to LC only patients, total leucocyte, neutrophil, and lymphocyte, levels were significantly higher in LC-COPD patients. No significant differences were found in levels of albumin, total proteins, fibrinogen, C-reactive protein (CRP), globular sedimentation (GSV), and body weight loss between the two study groups of patients.

### 2.2. Number and Area of TLSs and Number of GCs in Lung Samples

#### 2.2.1. Differences between LC-COPD and LC in Either Tumor or Non-Tumor Lung Samples

Both numbers of TLSs corrected by area (TLSs/mm^2^) and total area of TLSs (mm^2^) significantly decreased in the tumors of LC-COPD patients compared to LC group (Figure 1A–C). The number of GCs also significantly declined in LC-COPD patients compared to LC patients (Table 2 and Figure 1D).

#### 2.2.2. Differences between Tumor and Non-Tumor Lung Samples in LC-COPD and LC Patients

Compared to non-tumor specimens, both numbers and areas of TLSs were significantly higher in tumor lungs than in non-tumor lungs in both study groups (Figure 1A–C). The GCs number also significantly increased in lung tumors compared to non-tumor specimens in both study groups (Table 2 and Figure 1D).

### 2.3. T and B Cell Levels in Lung Samples

#### 2.3.1. Differences between LC-COPD and LC in either Tumor or Non-Tumor Lung Samples

Total numbers of T cells/μm^2^ and B cells/μm^2^ did not differ in either tumor or non-tumor specimens between LC-COPD and LC patients (Figure 2A–C).

#### 2.3.2. Differences between Tumor and Non-Tumor Lung Samples in LC-COPD and LC Patients

The number of T cells/μm^2^ was significantly lower in tumor samples compared to non-tumor lungs of both groups of patients (Figure 2A–C, top panel). Total numbers of B cells/μm^2^ were significantly greater in the tumors compared to non-tumor lungs in both LC and LC-COPD groups of patients (Figure 2A–C, bottom panel).

Among LC-COPD patients, statistically significant associations were seen between the percentage of T cells in lung tumor specimens and forced expiratory volume in one second/forced vital capacity (FEV_1_/FVC) (*R* = 0.228 and *p* = 0.032) and the percentage of B cells and FEV_1_/FVC (*R* = −0.370 and *p* < 0.001, Figure 2D).

### 2.4. Associations of TLSs with OS in LC and LC-COPD Patients

When all patients were analyzed together, a lower number of TLSs (cut-off: 1.944/mm^2^) was associated with a poorer 10-year survival (Figure 3A). When patients were subdivided according to the presence of COPD, no significant differences were observed between a low number of TLSs and survival (Figure 3B). As to the area of TLSs in the tumors (cut-off value: 1.112 mm^2^), a significantly worse survival was observed in the patients with lower levels of TLS area (Figure 3C). When patients were subdivided according to underlying COPD, smaller TLS areas in the tumors were also significantly associated with significantly poorer survival than those with greater areas of TLSs (Figure 3D). Moreover, when patients were stratified according to COPD severity (GOLD I and II stages), smaller areas of TLSs were also associated with a significantly poorer survival (Figure 3E). Interestingly, the presence of underlying COPD in this cohort was also significantly associated with a lower 10-year patients’ survival as shown in Figure 3F.

### 2.5. Associations of B and T Cells with Survival in LC and LC-COPD Patients

Patient 10-year survival was not significantly modified by the levels of T cells (cut-off: 38.08 cells × µm^−2^, %) in the tumors in either LC or LC-COPD patients (Figure 4A,B). When all patients were analyzed together, lower numbers of B cells in the tumors (cut-off: 90.80 cells × µm^−2^, %) were significantly associated with a significantly worse survival (Figure 4C). Interestingly, in patients with LC-COPD, but not in those with LC, lower levels of B cells in the tumors were significantly associated with poorer survival than in patients with higher levels of B cells (Figure 4D).

## 3. Discussion

In the current study, the main findings were that, in tumor specimens of patients with LC and underlying COPD, the numbers of TLSs and GCs were reduced. Smaller areas of TLSs and lower numbers of B cells were associated with a poorer 10-year survival of the patients, and this was also related to the severity of the COPD as measured by GOLD stage. Moreover, the presence of a chronic respiratory disease, such as COPD, per se, was also associated with a worse survival among all the patients with LC. The main results encountered in the study are discussed below.

TLSs are organized similarly to lymph nodes or spleen and their function in tissues is probably linked to underlying inflammation. In fact, TLSs are present in organs of chronic inflammatory diseases and are characterized by lymphoid genesis. TLSs are composed by large B cell follicles surrounded by T cells, which may contain dendritic cells [21]. Greater numbers of TLSs were detected in the small airways [22,23] and lungs [15,20,24] of patients with COPD and in the lungs of mice exposed to chronic cigarette smoke [24]. Furthermore, B-cell infiltration in TLSs was also shown to perpetuate inflammatory events in lung specimens that may lead to COPD progression in the patients [16].

The occurrence of TLSs has also been demonstrated in tumor samples of patients with NSCLC mainly characterized by the presence of follicular B cells, mature dendritic cells, and T cells [17,25]. In those studies, the density of mature dendritic cells was shown to correlate with better clinical outcomes in patients with early stages of NSCLC [17,25]. The same authors [19] also demonstrated that B cell density within the TLSs may also be a surrogate for the patients’ long-term survival in early stages of NSCLC, implying a role for B-cell mediated immunity in these patients.

In the current investigation, the numbers and area of TLSs were significantly reduced in lung tumor specimens of patients with COPD compared to those without this condition. Moreover, the numbers of GCs, sites of B cell proliferation and differentiation, were also significantly lower in the tumor samples of the COPD patients than in those without this disease. These results are in line with the decline in the number and area of the TLSs, implying that patients with underlying COPD may be less immunocompetent against tumorigenesis. Interestingly, the proportions of B cells were increased in the lung tumors of both groups of patients, with no significant differences between them, while T-cell counts within the TLSs declined in the tumor specimens, with no effect of COPD on those numbers. In fact, B cells were shown to have prognostic value regardless of the numbers of CD8+ T cells in tumors [26]. T cells may become exhausted in tumors including patients with COPD [27], which shows the existing correlations between immune checkpoints and TLSs [26]. These results are similar to those previously reported [27] in a retrospective study of patients with LC, in which a significant proportion of the patients were also COPD. Nonetheless, these results are somehow counter to previous reports in which the number of several types of T cells were increased in tumors of patients with COPD [28]. The methodologies (immunohistochemistry versus flow cytometry) employed in each study, the approaches (prospective versus retrospective cohorts of patients) used in each case, and the degree of the airway obstruction may account for discrepancies among investigations [27,28]. Importantly, the proportions of B cells within the TLSs were greater in the tumor samples than in the tumor-free parenchyma. These findings are in agreement with those previously shown in tumor tissues (sarcoma and melanoma), in which a great amount of B cells was also identified in patients [26,29,30].

In the study, all of the patients from the Lung Cancer Mar Cohort were prospectively followed up to ten years. When all LC patients were analyzed together, a reduced number of TLSs in the lung tumor specimens was associated with a poorer survival in the patients compared to those with greater numbers of TLSs. Furthermore, a smaller area of TLSs in the lung tumors was also associated with a worse prognosis, especially in the patients with underlying COPD. Likewise, a low proportion of B cells in the lung tumors also correlated with a poorer survival among patients with COPD. These are relevant findings that are line with recent reports [26,29,30], in which B cell-enriched TLSs were the best prognostic factor among study patients regardless of the proportions of T cells.

Importantly, in those seminal investigations [26,29,30], the presence of TLSs and B cells was associated with greater survival rates, as well with improved response to immunotherapy in patients with either melanoma [29,30] or sarcoma [26]. In accordance, findings encountered in the present study also demonstrated that greater areas of TLSs and of B cell proportions led to better survival rates in patients with LC that were followed up for a long period of time (10 years). Interestingly, these associations were especially blatant in patients with underlying COPD. In fact, reduced area of TLSs was also significantly associated with a poorer survival when COPD patients were analyzed independently according to the severity of their disease as identified by GOLD stages I and II (91% of all the patients). In addition, patients with underlying COPD were also those who died significantly earlier than patients with no COPD (Figure 3F, hazard ratio: 2.35). Indeed, these are confirmatory findings of what had already been published in previous investigations [31,32,33], showing that mortality rates were significantly higher in LC patients with underlying COPD [34,35].

Study Limitations

A potential study limitation is related to the relatively reduced number of patients for the amount of variables and subgroups that were analyzed. Nonetheless, the study hypothesis was confirmed. Additionally, the number of patients was correct, according to the estimations made using statistical analyses as described in Methods. The specific role of cigarette smoking was not assessed in the study, despite that its burden was significantly higher in the LC-COPD patients. However, in a multivariate analysis, in which the variable packs–year was also included, no significant differences were observed between the two groups of patients. These results are in line with those showed by Mark et al. [28], who reported no significant effects of cigarette smoking on Th1 cell profile in lung tumors.

## 4. Materials and Methods

### 4.1. Study Design and Ethics

This is a cross-sectional, prospective study designed following the World Medical Association guidelines (seventh revision of the Declaration of Helsinki, Fortaleza, Brazil, 2013) [36] for research on human beings and approved by the institutional Ethics Committee on Human Investigation (protocol # 2008/3390/I, 4 February, 2008, Hospital del Mar–Instituto Hospital del Mar de Investigaciones Médicas, Barcelona, Spain). All patients invited to participate in the study signed the written informed consent.

Patients were prospectively recruited from the Lung Cancer Clinic of the Lung Cancer Unit at Hospital del Mar (Barcelona, Spain). All of the patients were part of the Lung Cancer Mar Cohort that started in 2008. The last patients were enrolled in March 2018. For this observational study, 133 patients with LC were recruited. Candidates for tumor resection underwent pulmonary surgery prior to administration of any sort of adjuvant therapy. Specimens from the tumor and non-tumor lungs were collected from all the study subjects. Patients were further subdivided post-hoc into two groups on the basis of underlying COPD: 1) 90 patients with LC and COPD (LC-COPD group), and 2) 43 patients with LC without COPD (LC group).

LC diagnosis and staging were established by histological confirmation and classified according to currently available guidelines for the diagnosis and management of LC [37,38]. Tumor, node, and metastasis (TNM) staging was defined as stated in the 8th edition of the Lung Cancer Stage Classification [39]. In all cases, pre-operative staging was performed using chest and upper abdomen Computed Tomography (CT) scan and Fluoro-deoxy-glucose positron emission tomography/computed tomography (PET) body-scan. When suspected mediastinal lymph-node involvement, a fiber optic bronchoscopy with endo-bronchial ultra-sound (EBUS), and trans-tracheal biopsy of the suspected nodes were performed. In case of negative results, a surgical exploration of the mediastinum: cervical video-assisted mediastinal lymphadenectomy (VAMLA) and/or anterior mediastinotomy were performed, the latter depending on the location of the suspected nodes. Notwithstanding, in all surgical cases, intra-operative systematic hilar and mediastinal lymphadenectomy (at least, ipsilateral paratracheal, subcarinal, and ipsilateral pulmonary ligament) was performed as previously recommended [40,41]. Standard clinical guidelines were used to establish the selection of patients and contraindications for thoracic surgery as previously described [42]. Decisions on the best therapeutic approach were always made during the weekly meetings of the Multidisciplinary Lung Cancer Committee. Lung tumor resections were applied using classical thoracotomy for all the patients in this study. In the present study, exclusion criteria were: small cell lung cancer (SCLC), severe malnutrition status, chronic cardiovascular disease, metabolic or clot system disorders, signs of severe inflammation and/or bronchial infection (bronchoscopy), current or recent invasive mechanical ventilation, or long-term oxygen therapy. The presence/absence of these diseases was confirmed using standard clinical tests: exercise capacity electrocardiogram, clinical examination, blood tests, bronchoscopy, and echocardiography.

### 4.2. Clinical Assessment

In all patients, lung function parameters were assessed following standard procedures. Diagnosis and severity of patients with COPD were determined according to current guidelines [5,43]. Nutritional evaluation included the assessment of body mass index (BMI) and nutritional blood parameters from all patients.

### 4.3. Sample Collection and Preservation

Lung samples were obtained from tumors and the surrounding non-tumor parenchyma following standard technical procedures during thoracotomy for the standard care in the treatment of lung tumors. In all patients, the expert pulmonary pathologist selected tumor and non-tumor lung specimens of approximately 10 × 10 mm^2^ area from the fresh samples. Non-tumor specimens were collected as far distal to the tumor margins as possible (average >7 cm). Fragments of both tumor and non-tumor specimens were fixed in formalin and embedded in paraffin blocks until further use.

### 4.4. Identification of B Cells, T Cells, and TLSs in the Lung Specimens

B cells, T cells, and TLSs were identified on three-micrometer lung tumor and non-tumor cross-sections using double-staining immunohistochemical procedures (EnVision DuoFLEX Doublestain System, Dako North America Inc., Carpinteria, CA, USA) following the manufacturer’s instructions and previous study [10,44,45]. B and T cells were identified by staining of the lung samples with specific antibodies for B cells (anti-CD20 antibody, clone L26, Dako) and T cells (anti-CD3 antibody, Dako). Following deparaffinization, lung sample cross-sections were immersed in preheated antigen-retrieval solution (Dako high pH solution) at 95 °C for 20 min to be then allowed to cool down to room temperature. Slides were washed several times with wash buffer (Dako wash buffer solution). Endogenous peroxidase activity was blocked for 15 min with Dako endogenous enzyme blocking agent. Samples were incubated with anti-human CD3 rabbit polyclonal primary antibody for 40 min. The second incubation was performed for one hour with anti-human CD20 mouse monoclonal antibody. Dextran polymer (EnVision DuoFLEX, Dako) was used as the secondary antibody. Samples were subsequently incubated for 20 min with horseradish peroxidase for mouse monoclonal (CD20) and alkaline phosphatase for rabbit polyclonal (CD3) antibodies. Slides were gently washed and incubated for 10 min with diaminobenzidine (EnVision DuoFLEX, 3,3′-Diaminobenzidine) as a chromogen for the mouse monoclonal antibody (brown reaction product; anti-CD20 antibody) and with liquid permanent red (EnVision DuoFLEX LPR) as a chromogen for the rabbit polyclonal antibody (red reaction product; anti-CD3 antibody).

All procedures were conducted at room temperature. Hematoxylin counterstaining was performed for two minutes, and slides were mounted for conventional microscopy. Images were taken under a light microscope (Olympus, Series BX50F3, Olympus Optical Co., Hamburg, Germany) coupled with an image-digitizing camera (Pixera Studio, version 1.0.4, Pixera Corporation, Los Gatos, CA, USA). The number of cells and total area (μm^2^) were measured in each of the lung specimens (both tumor and non-tumor samples) using the Image J software (National Institute of Health, Maryland, MD, USA).

In each lung section, the total amount of B cells (CD20-positively-stained) and T cells (CD3-positively-stained) were quantified blindly by two independent observers who were previously trained for that purpose. Data are presented as the percentage of either B or T cells separately in the measured area in both tumor and non-tumor lung specimens (% B cells/μm^2^ and % T cells/μm^2^, respectively).

Numbers of TLSs were also manually counted by two independent trained observers after identification of the cell types (B and T cells) that composed these structures using Image J software (National Institute of Health). In addition, total area (mm^2^) of each TLSs was also measured in both tumor and non-tumor specimens using Image J software. Data are presented as the number of TLSs in the measured area in both tumor and non-tumor samples (number of TLSs/mm^2^) and as the mean area of all the identified and counted TLSs (mm^2^).

### 4.5. Identification of GCs in TLSs of Lung Specimens

In a subgroup of patients (*n* = 61), the presence of GCs within the TLSs was also specifically evaluated in each lung tumor and non-tumor specimens on three-micrometer sections using hematoxylin and eosin staining by two independent observers [10,46,47]. Images of the stained lung sections (tumor and non-tumor) were captured with a light microscope (Olympus, Series BX50F3, Olympus Optical Co., Hamburg, Germany) coupled with an image-digitizing camera (Pixera Studio, version 1.0.4, Pixera Corporation, Los Gatos, CA, USA). GCs were selected by the presence of two separate topographic zones: 1) one dark-stained area, which was characterized by a dense population of lymphocytes, and 2) a light-stained area, which was characterized by a low-density lymphocyte site. Data are expressed as the number of GCs in all study groups of patients.

### 4.6. Statistical Analyses

The normality of the study variables was examined using the Shapiro-Wilk test. For an initial descriptive analysis of clinical parameters, qualitative variables were described as frequencies (number and percentage) and quantitative variables as mean and standard deviation. Differences between LC and LC-COPD were assessed using Student’s t-test or Mann-Whitney U tests for parametric and non-parametric variables, respectively. Chi-square test was used to assess differences between the two groups for the categorical variables.

Differences among the different biological variables were explored using the Kruskal-Wallis equality-of-populations rank test, followed by Dunn’s Pairwise Comparison test (Sidák adjustment) for the two sample types and patient groups.

OS was defined as the time from the date of diagnosis of LC to the date of death from this disease or the last follow-up, which was completed in December 2018. The median follow-up duration was 37.9 months (P25 = 20.0 months, P75 = 65.4 months). Patients were followed up to a maximum period of 10 years. Patients who did not died of lung cancer were excluded in the investigation.

Threshold analysis was carried out for each continuous biological variable to determine the best cut-off point as predictor of OS, which was the endpoint in the study. The cut-off point was defined using the web-based software Cutoff Finder [48], which has also been previously used in other studies [49,50]. For each biological variable, we identified the threshold level at which a log-rank test allowed segregation of patients into groups with better and worse survival.

Moreover, taking each variable categorized into two groups, estimated power for two-sample comparisons of survivor functions Log-rank test was applied using the Freedman method. Accepting an alpha risk of 0.05 in a two-sided test with 87 and 38 patients in each group (post hoc subdivision), the statistical power was 100% (both number and area of TLSs), T cells (86%), and 100% (B cells). Kaplan-Meier survival curves were performed for each dichotomized variable (below versus above cutoff values, described as Lo and Hi) and log-rank test *p*-value was estimated. Pearson’s correlation analyses were performed to explore potential correlations between clinical and biological variables. Statistical significance was established at *p* ≤ 0.05. All statistical analyses were carried out using the software Stata/MP 15 (StataCorp LLC, Texas, TX, USA).

## 5. Conclusions

A decline in the surface and numbers of TLSs was observed in lung tumors of patients with underlying COPD, which was significantly associated with a poorer survival in these patients. An increase in B cell proportions was seen within the TLSs in tumors of LC patients with and without chronic respiratory disease, and in the latter group, lower levels of B cells correlated with lower survival. The immune tumor microenvironment differs in patients with underlying COPD and these different phenotypes may eventually impact the response to immunotherapy in patients with LC. Thus, the presence of underlying respiratory conditions should be targeted when designing immune therapeutic strategies in LC.

## Figures and Tables

**Figure 1 cancers-12-02644-f001:**
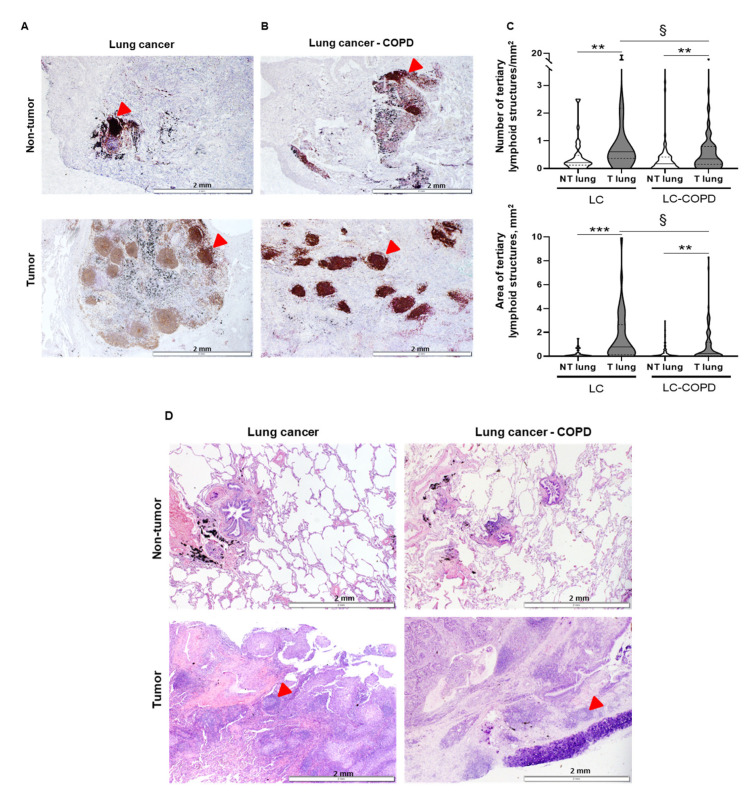
TLSs and germinal centers in tumor and non-tumor lungs of patients. (**A,B**) Representative examples of double immunohistochemical staining for TLSs indicated by red arrows in lung cancer (LC) and LC-COPD patients, respectively. (**C**) Violin plot with median (continuous line) and interquartile range (discontinuous line) of the number of TLSs corrected by area (number/mm^2^, top panel) and total area of TLSs (mm^2^, bottom panel), respectively. Comparisons were made between the non-tumor (NT) and tumor (T) samples, and the LC and LC-COPD groups of patients. Statistical significance: **, *p* ≤ 0.01; ***, *p* ≤ 0.001 between tumor and non-tumor lungs in either LC or LC-COPD patients, §, *p* ≤ 0.05 in tumor samples between LC and LC-COPD patient groups. (**D**) Representative examples of hematoxylin and eosin staining for the germinal centers contained within the TLSs in LC and LC-COPD patients. Red arrows point towards germinal centers. Definition of abbreviations: TLSs, tertiary lymphoid structures.

**Figure 2 cancers-12-02644-f002:**
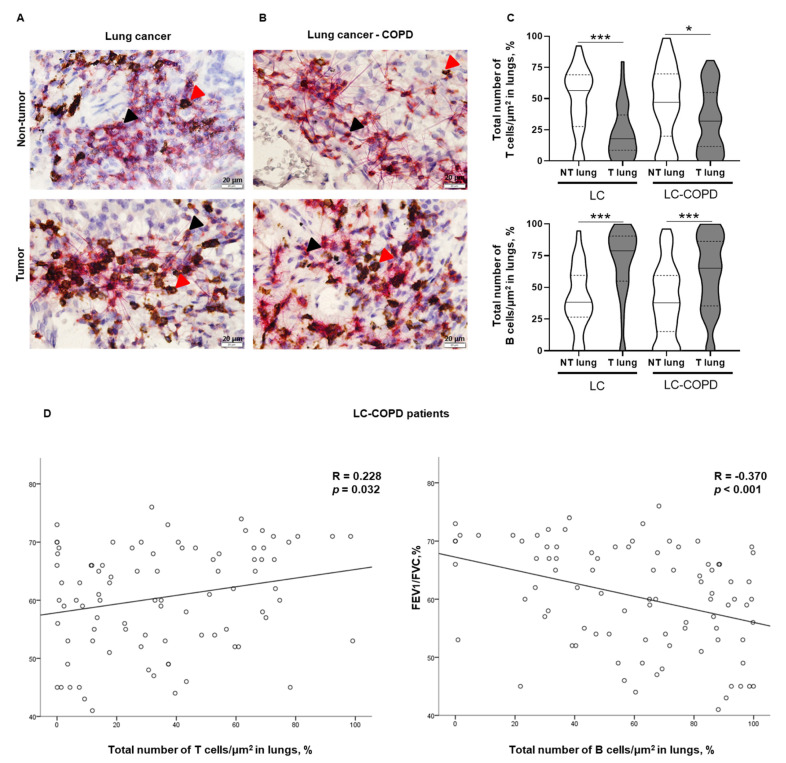
T and B cell counts in tumor and non-tumor lungs of patients and their correlations with lung function parameters. (**A**,**B**) Representative double immunohistochemical staining sections of T and B cells in non-tumor and tumor lung specimens of LC and LC-COPD patients, respectively. T cells (CD3+) indicated by black arrows were stained in red color, and B cells (CD20+) indicated by red arrows were stained in brown color, respectively. (**C**) Violin plot with median (continuous line) and interquartile range (discontinuous line) of the number of T cells (top panel) and B cells (bottom panel) as indicated by the percentage of T and B cells in the total measured area respectively. Black stained regions within the lungs correspond to anthracosis. (**D**) Statistically significant correlations between FEV_1_/FVC and T cell numbers (positive), and B cell numbers (inverse) in LC-COPD patients. Definition of abbreviations: LC, lung cancer; COPD, chronic obstructive pulmonary disease; CD, cluster of differentiation; FEV_1_, forced expiratory volume in one second; FVC, forced vital capacity. Statistical significance: ***, *p* ≤ 0.001, * *p* < 0.05 between tumor and non-tumor samples in both study groups.

**Figure 3 cancers-12-02644-f003:**
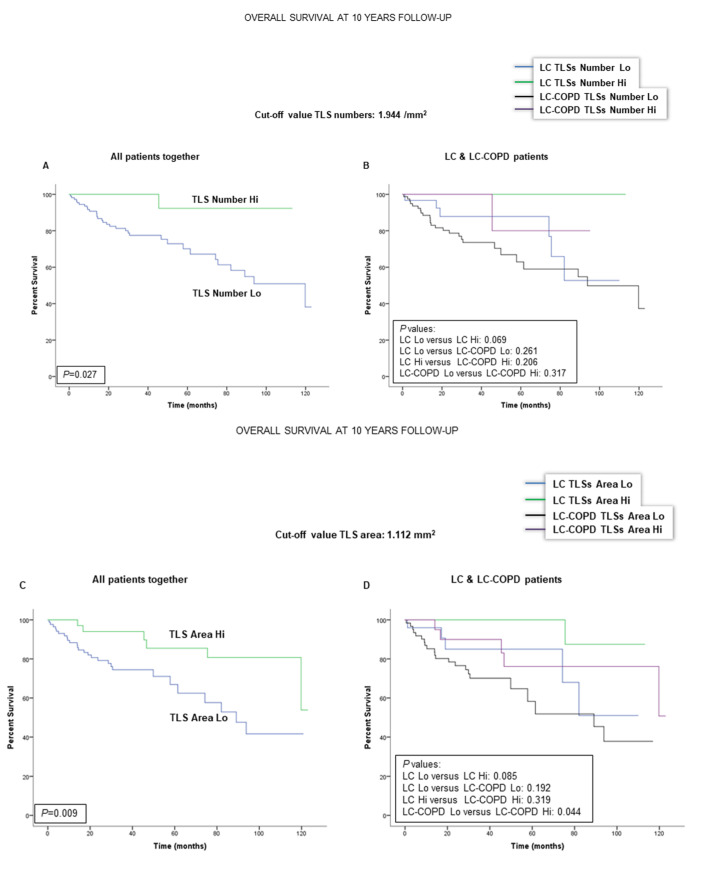
Kaplan-Meier survival curves of the two groups of patients according to TLS levels in tumors. (**A**) Kaplan-Meier survival curves for overall survival (OS) in all patients based on the cut-off value of the number of TLSs (above and below the cut-off value: 1.944/mm^2^). Patients with lower numbers had a significantly worse survival. (**B**) Kaplan-Meier survival curves for OS in LC patients with and without COPD based on the cut-off value of the number of TLSs (above and below the cut-off value: 1.944/mm^2^). No significant differences were detected. (**C**) Kaplan-Meier survival curves for OS in all patients based on the cut-off value of the total area of TLSs (above and below the cut-off value: 1.112 mm^2^). Patients with smaller areas of TLSs had a significantly worse survival. (**D**) Kaplan-Meier survival curves for OS in LC patients with and without COPD based on the total area of TLSs (above and below the cut-off value: 1.112 mm^2^). Smaller areas of TLSs were significantly associated with poorer survival among LC-COPD patients. (**E**) Kaplan-Meier survival curves for OS in LC-COPD patients with GOLD I-II stages based on the total area of the TLSs (above and below the cut-off value: 1.112 mm^2^). Patients with smaller areas of TLSs had a significantly worse survival. (**F**) Kaplan-Meier survival curves for OS in LC patients according to the presence of underlying COPD. COPD per se was associated with a worse survival among the study patients. Definition of abbreviations: LC, lung cancer; COPD, chronic obstructive pulmonary disease; Hi, high level (above cut-off value); Lo, low level (below cut-off value); TLS, tertiary lymphoid structures; GOLD, global initiative for chronic obstructive lung disease.

**Figure 4 cancers-12-02644-f004:**
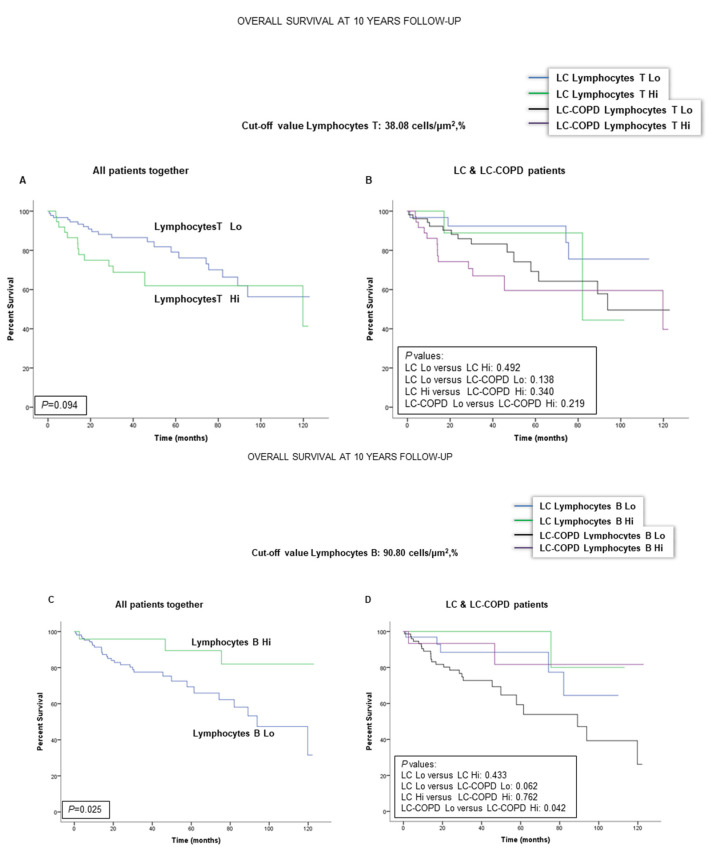
Kaplan-Meier survival curves of the two groups of patients according to T and B cell counts in tumors. (**A**) Kaplan-Meier survival curves for OS in all patients based on the percentage of T cell lymphocytes (above and below the cut-off value: 38.08 cells/μm^2^, %). (**B**) Kaplan-Meier survival curves for OS in LC patients with and without COPD based on the percentage of T lymphocytes (above and below the cut-off value: 38.08 cells/μm^2^, %). No significant differences were observed in any of the patient groups according to the cut-off value of the percentage of T cells within the TLSs. (**C**) Kaplan-Meier survival curves for OS in all patients based on the percentage of B cell lymphocytes (above and below the cut-off value: 90.80 cells/μm^2^, %). Patients with lower proportions of B cells had a significantly worse survival. (**D**) Kaplan-Meier survival curves for OS in LC patients with and without COPD based on the percentage of B lymphocytes (above and below the cut-off value: 90.80 cells/μm^2^, %). Smaller proportions of B cells were significantly associated with poorer survival among LC-COPD patients. Definition of abbreviations LC, lung cancer; COPD, chronic obstructive pulmonary disease; Hi, high level (above cut-off value); Lo, low level (below cut-off value).

**Table 1 cancers-12-02644-t001:** Clinical and functional characteristics of the study patients.

Anthropometric Variables	Lung Cancer (*n* = 43)	Lung Cancer-COPD (*n* = 90)
Age, years	65 (12)	67 (8)
Male, N/Female, *N*	17/26	78/12 ***
BMI, kg/m^2^	27 (4)	26 (4)
Smoking history		
Current: *N*, %	13, 30	43, 48
Ex-smoker: *N*, %	8, 19	44, 49 **
Never smoker: *N*, %	22, 51	3, 3 ***
Pack-years	17 (22)	56 (25) ***
Lung function parameters		
FEV_1_	90 (12)	67 (15) ***
FEV_1_/FVC, %	75 (6)	61 (9) ***
DLco, %	85 (14)	67 (18) ***
Kco, %	85 (12)	69 (17) ***
GOLD Stage		
GOLD Stage I: *N*, %	NA	19, 21
GOLD Stage II: *N*, %	NA	63, 70
GOLD Stage III: *N*, %	NA	8, 9
TNM staging		
Stage 0–II: *N*, %	37, 86	73, 81.1
Stage III: *N*, %	6, 14	13, 14.5
Stage IV: *N*, %	0, 0	4, 4.4
Histological diagnosis		
Squamous cell carcinoma: *N*, %	5, 12	16, 17.8
Adenocarcinoma: *N*, %	32, 74	68, 75.6
Others: *N*, %	6, 14	6, 6.7
Blood parameters		
Total leucocytes/μL	7.39 (2.42) × 10^3^	9.17 (2.93) × 10^3^ ***
Total neutrophils/μL	4.82 (2.49) × 10^3^	6.01 (2.61) × 10^3^ **
Total lymphocytes/μL	1.76 (0.78) × 10^3^	2.32 (1.61) × 10^3^ *
Albumin (g/dL)	4.3 (0.4)	4.1 (0.6)
Total proteins (g/dL)	7.0 (0.6)	6.8 (0.8)
Fibrinogen (mg/dL)	420 (130)	454 (151)
CRP (mg/dL)	6.5 (8.3)	7.5 (13.1)
GSV (mm/h)	27 (14)	27 (16)
Body weight loss, kg		
0, *N*, %	40, 93	82, 91
1–5, *N*, %	1, 2	3, 3
6–10, *N*, %	2, 5	5, 6

Continuous variables are shown as mean and standard deviation, while categorical variables are described as the number of patients in each group and the percentage in the study group with respect to the total population. Definition of abbreviations: N, number; kg, kilograms; m, meters; BMI, body mass index; FEV_1_, forced expiratory volume in one second; FVC, forced vital capacity; DL_CO_, carbon monoxide transfer; K_CO_, Krogh transfer factor; GOLD: Global Initiative for Chronic Obstructive Pulmonary Disease; NA, not applicable; TNM, tumor, nodes, metastasis; CRP, C-reactive protein; GSV, globular sedimentation velocity; L, liter; COPD, chronic obstructive pulmonary disease. Statistical analyses and significance: * *p* < 0.05, ** *p* < 0.01, *** *p* < 0.001 between LC-COPD patients and LC patients.

**Table 2 cancers-12-02644-t002:** Number of germinal centers within tertiary lymphoid structures.

Germinal Centers	Lung Cancer (*n* = 18)		Lung Cancer-COPD (*n* = 43)	
	NT Lung	T Lung	NT Lung	T Lung
0, *n* (%)	17 (94)	10 (56) *	43 (100)	36 (84) **^,§^
>1, *n* (%)	1 (6)	8 (44) *	0 (0)	7 (16) **^,§^

Values are represented as number and percentage of the total samples in both tumor (T) and non-tumor (NT) samples in both LC and LC-COPD groups of patients. Statistical analyses and significance: * *p* < 0.05, ** *p* < 0.01 between tumor and non-tumor lung specimens in either LC or LC-COPD groups of patients, § *p* < 0.05 in tumor lung specimens between LC and LC-COPD patients. The digit 0 means absence of germinal centers (GCs) in the samples.

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
