# Peer review of "B Cells and Tertiary Lymphoid Structures Influence Survival in Lung Cancer Patients with Resectable Tumors"

_cancers, 2020, doi:10.3390/cancers12092644_

Round 1

Reviewer 1 Report

The Authors reported an analysis on the role of tertiary lymphoid structures (TLSs) on survival of 133 lung cancer patients with resectable tumors recruited at Hospital Del Mar, Barcelona, from 2008. The main finding is that in LC-COPD patients greater area of TLSs and proportion of B cells were
associated with longer survival rates.The results are interesting, but the number of patients is overall low, and the Authors should highlight this weakness in the discussion. Moreover:

1) Table 1 should be improved in the quality of presentation

2) The study started in 2008: when it was enrolled the last patients?

3) The subgroup analyses presented are too many, particular for LC and LC-COPD patients (see figure 3B, 3D, 4B and 4D).

4) Also the analysis on the role on adjuvant chemotherapy could be affected by the low number of patients and I think that it should be deleted

Author Response

REVIEWER #1:

REVIEWER #1:

C1

Comments and Suggestions for Authors

The Authors reported an analysis on the role of tertiary lymphoid structures (TLSs) on survival of 133 lung cancer patients with resectable tumors recruited at Hospital Del Mar, Barcelona, from 2008. The main finding is that in LC-COPD patients greater area of TLSs and proportion of B cells were associated with longer survival rates.The results are interesting, but the number of patients is overall low, and the Authors should highlight this weakness in the discussion. Moreover:

R1

We thank the reviewer for having reviewed our manuscript and for the insightful comments. We are also grateful to the reviewer for having considered that the results of our study are interesting. We have modified the entire manuscript on the basis of the concerns raised by the two reviewers and have also provided responses to each comment in this document. We have modified the Discussion and have included a Study limitations section in the revised manuscript, in which the relatively low number of patients and other potential weaknesses have been discussed in the revised manuscript. We invite the reviewer to read the revised text in lines 305-314.

Moreover, the estimation of the number of patients required to meet the study objectives has also been explained in the Methods section, under the Statistical analyses subheading in the revised manuscript version (See lines 439-443).

C2

1) Table 1 should be improved in the quality of presentation

R2

We thank the reviewer for this comment. The variables corresponding to adjuvant therapy have been omitted in the revised Table 1. The format of this Table has improved to make it fit within one single page. This is the Journal’s format for original research articles.

C3

2) The study started in 2008: when it was enrolled the last patients?

R3

We thank the reviewer for this comment. The last patients were enrolled in March 2018 (See line 331 in the revised manuscript).

C4

3) The subgroup analyses presented are too many, particular for LC and LC-COPD patients (see figure 3B, 3D, 4B and 4D).

R4

We thank the reviewer for this comment. We decided to leave the subanalyses in LC and LC-COPD patients as the investigation of the impact of COPD on LC is one of the main objectives of the current study. Moreover, as the calculations of the number of patients have shown that 38 and 87 patients in the groups LC and LC-COPD, respectively were sufficient to confirm the study hypothesis which particularly focused on the effects of COPD condition on overall survival (See lines 439-443).

C5

4) Also the analysis on the role on adjuvant chemotherapy could be affected by the low number of patients and I think that it should be deleted.

R5

We thank the reviewer for this comment. The analysis of adjuvant therapy has been deleted in the revised manuscript (See Table 1, and the Results and Discussion sections in the revised manuscript).

Reviewer 2 Report

The authors sowed that TLS, GC, and B cells in the tissues of lung cancer patients with COPD were decreased and the prognosis was poor. In COPD patients, the immune microenvironment is different from normal lung cancer and they indicated the influence the efficacy of immunotherapy.

1.Although there are many reports that increased TLS leads to increased T cells, this study shows that T cell infiltration does not affect prognosis and B cells have an effect in LC-COPD. What do you think is the reason for that?

2.Although most COPD cases are smokers, what do you think is the relationship between decreased TLS and decreased GC and B cells?

3.Is there any evidence that immunity is fundamentally suppressed in COPD cases?

Minor points:

1.In the Methods section, it is described as counting TLS, but Image J is also used. It is unclear whether the area or number of TLS per unit area of tumor is being evaluated.

2.The identification of GC is ambiguous. The procedure section states that the GC was determined by HE staining, while the Figure Legend indicates that immunohistochemistry was performed. Immunohistochemical staining should be evaluated with CD 23 or BCL6.

3.In Figure 2D, there was a significant correlation, but the R value was 0.228, so it cannot be said to be a significant correlation.

4.The double-staining picture of T and B cells in Figure 2A is not clear. I need a replacement.

Author Response

REVIEWER #2:

C1

Comments and Suggestions for Authors

The authors sowed that TLS, GC, and B cells in the tissues of lung cancer patients with COPD were decreased and the prognosis was poor. In COPD patients, the immune microenvironment is different from normal lung cancer and they indicated the influence the efficacy of immunotherapy.

R1

We thank the reviewer for having reviewed our manuscript and for the insightful comments. We have modified the entire manuscript on the basis of the concerns raised by the two reviewers and have also provided responses to each comment in this document.

C2

  1. Although there are many reports that increased TLS leads to increased T cells, this study shows that T cell infiltration does not affect prognosis and B cells have an effect in LC-COPD. What do you think is the reason for that?

R2

We thank the reviewer for this comment. Studies have demonstrated that in chronic inflammation and cancer conditions, the infiltrating T cell function is impaired by the expression of immune-checkpoints such as PD-1, PD-L1, CTLA-4, and TIM-3, namely T cell exhaustion. Furthermore, other components of TLSs in the tumor microenvironment such as B cells and GCs were also shown to be associated with better prognosis in different cancer types including LC (See references 28-30 of the revised manuscript). In the revised Discussion these aspects have been discussed (See lines 270-272).

C3

  1. Although most COPD cases are smokers, what do you think is the relationship between decreased TLS and decreased GC and B cells?

R3

We thank the reviewer for this comment. We did not assess the potential influence of cigarette smoking in the current study. The proportions of current and ex-smokers were greater in the LC-COPD patients than in the LC group. However, we do not believe that this has had any significant effect on the study results as TLSs, GCs, and B cell levels were higher in lung tumors compared to non-tumor specimens in both groups of patients: LC and LC-COPD patients. In addition, smoking status did not influence Th1 or Treg levels in lung tumors in a previous study (See reference 28). Moreover, the multivariant analyses in which the variable packs-year was included, no significant differences (P=0.154) were identified for this variable in the present study (See also lines 309-314 in the revised manuscript).

C4

  1. Is there any evidence that immunity is fundamentally suppressed in COPD cases?

R4

We thank the reviewer for this comment. In view of the current findings it is possible to conclude that tumors of patients with underlying COPD experience a significant decrease in TLSs numbers and areas as well as in GCs. However, the reduction in T cell levels was similar in tumors of both groups of patients with underlying COPD. Thus, it will be possible to conclude that immunity involving TLSs and GCs was somehow suppressed in tumors of COPD patients. These aspects were already explained in the Discussion section and have also been kept the same in the revised manuscript version (See lines 262-267).

Studies have shown that in chronic inflammation and cancer conditions, T cells are exhausted by the expression of immune checkpoints such as PD1, PD-L1, and CTLA-4. Moreover, Biton et al reported that the tumor-infiltrating lymphocytes were more exhausted in lung tumors of LC-COPD compared to LC-only patients (See reference 27 in the revised manuscript). In the same study, patients were treated with an anti-PD-1 antibody (Nivolumab), and LC-COPD patients had significantly longer progression-free survival rates compared to LC-only patients. Mark et al. also demonstrated that the presence of COPD in LC patients was associated with improved response to immune checkpoint inhibitors (anti-PD-1 or anti-PD-L1 therapy, see reference 28 of the revised manuscript). These aspects have been included in the revised Discussion section (See lines 262-282).

C5

Minor points:

1.In the Methods section, it is described as counting TLS, but Image J is also used. It is unclear whether the area or number of TLS per unit area of tumor is being evaluated.

R5

We thank the reviewer for this comment. Both area and number of TLSs per unit area of tumor were evaluated in the study. Data were presented as the number of TLSs in the measured area and as the areas of TLSs in each group (See Figures 1A-C and in Methods and Results sections (See lines 403-408 and line 119, respectively), and legend to Figure 1 in lines 132-133 in the revised manuscript).

C6

2.The identification of GC is ambiguous. The procedure section states that the GC was determined by HE staining, while the Figure Legend indicates that immunohistochemistry was performed. Immunohistochemical staining should be evaluated with CD 23 or BCL6.

R6

We thank the reviewer for this comment. GCs were identified using HE staining, the legend to Figure 1D has been corrected in the revised manuscript (See lines 137-138). In the Methods section, these aspects were well described and have been kept the same in the revised manuscript (See lines 409-418).

C7

3.In Figure 2D, there was a significant correlation, but the R value was 0.228, so it cannot be said to be a significant correlation.

R7

We thank the reviewer for this comment. It should be acknowledged that the correlation was statistically significant (P=0.032) even if the R value was relatively low. Thus, it is possible to conclude that this correlation was significant despite being of questionable clinical/biological relevance. The word “statistically” significant has been added to the corresponding sentence in the revised Results section and legend to Figure 2D (See lines 157 and 170, respectively).

C8

4.The double-staining picture of T and B cells in Figure 2A is not clear. I need a replacement.

R8

We thank the reviewer for this comment. Figure 2A has been replaced with another picture in the revised manuscript (See Figure 2A and line 161).

Round 2

Reviewer 1 Report

The authors reviewed the manuscript that is now acceptable in this form.

Reviewer 2 Report

The explanation of the relationship between smoking and the decrease in GC is somewhat insufficient, but if immunohistological analysis is not possible, it is considered unavoidable and accepted.

This manuscript is a resubmission of an earlier submission. The following is a list of the peer review reports and author responses from that submission.